# Intersection of Migration and Access to Health Care: Experiences and Perceptions of Female Economic Migrants in Canada

**DOI:** 10.3390/ijerph17103682

**Published:** 2020-05-23

**Authors:** Solina Richter, Helen Vallianatos, Jacqueline Green, Chioma Obuekwe

**Affiliations:** 1Faculty of Nursing, University of Alberta, Edmonton Clinic Health Academy, 11405, 87th Avenue, Edmonton, AB T6G 1C9, Canada; cobuekwe@ualberta.ca; 2Department of Anthropology, University of Alberta, 13-15 Tory Building, Edmonton, AB T6G2H4, Canada; vallianatos@ualberta.ca (H.V.); jgreen@ualberta.ca (J.G.)

**Keywords:** access, economic, experiences, female, health care, intersection, migrants

## Abstract

More people are migrating than ever before. There are an estimated 1 billion migrants globally—of whom, 258 million are international migrants and 763 million are internal migrants. Almost half of these migrants are women, and most are of reproductive age. Female migration has increased. The socioeconomic contexts of women migrants need investigation to better understand how migration intersects with accessing health care. We employed a focused ethnography design. We recruited 29 women from three African countries: Ghana, Nigeria, and South Africa. We used purposive and convenient sampling techniques and collected data using face-to-face interviews. Interviews were audio-recorded and transcribed verbatim. Data were analyzed with the support of ATLAS.ti 8 Windows (ATLAS.ti Scientific Software Development GmbH), a computer-based qualitative software for data management. We interviewed 10 women from both South Africa and Ghana and nine women from Nigeria. Their ages ranged between 24 and 64 years. The four themes that developed included social connectedness to navigate access to care, the influence of place of origin on access to care, experiences of financial accessibility, and historical and cultural orientation to accessing health care. It was clear that theses factors affected economic migrant women’s access to health care after migration. Canada has a universal health care system but multiple research studies have documented that migrants have significant barriers to accessing health care. Most migrants indeed arrive in Canada from a health care system that is very different than their country of origin. Access to health care is one of the most important social determinants of health.

## 1. Introduction

More people are migrating than ever before. There are an estimated 1 billion migrants globally—of whom, 258 million are international migrants and 763 million are internal migrants—one in seven of the world’s population [1]. Evidence suggests that migrants are a vulnerable group irrespective of the reasons for migration. Migration involves situational transition and exposes migrants to different levels of vulnerability [2]. This rapid increase in population movement has important public health implications and needs an adequate response from the health sector and an action plan to ensure access to health care for these vulnerable populations. The World Health Organization (WHO) Constitution of 1948 promotes the right of all to achieve a high standard of physical and mental health. Endorsed international human rights standards exist to protect the right to health of migrants. However, many migrants often lack access to health services and financial protection for health [1]. Migrant populations are likely to have unmet health needs due, at least in part, to linguistic and cultural barriers that contribute to challenges to accessing health care.

Almost half of these migrants are women, and most are of reproductive age [3]. There has been an increase in female migration, commonly referred to as the feminization of migration. Women are recently more involved in independent migration, mainly due to an increased demand for women in the care professions [4]. Over half of migrants to Canada (52%) were women in 2006 [5]. Migration frequently contributes to greater autonomy, human capital, and self-esteem in women [6]. “In some cases, the health of female migrants is improved via integration into better health systems in the host country. More often, however, the health of female migrants is affected negatively” [7]. International conventions like the Convention on the Elimination of Discrimination Against Women (CEDAW) stipulate that gender disparities against migrant women require attention, including access to life-saving and health-protective care [8,9].

Migration is a central part of Africa’s history and is high on the agenda of many African governments. The global discourse on migration centers around the recovery of the losses of workforce in the countries of origin, but families and individuals view it as a means to improve their family’s economic well-being [4] and, additionally, their human and social capital. A significant trend in migration patterns in Africa is the increased numbers of educated women who migrate independently to fulfill their individual economic needs [10]. Migration can be empowering for women through improved access to employment options and education and can potentially strengthen agency, “the ability to make independent decisions to achieve desired outcomes” [6]. Women migrants are increasingly part of the Canadian labour market; Canada’s female labour population increased by 9.5% and the employment rate of immigrant women was double that of Canadian-born women. However, increased labour participation can be complicated by family structure and other sociocultural factors. Some migrant mothers face challenges negotiating their extended families’ and husband’s nutritional and health adjustments postsettlement [11], while others face different stressors as single parents—and single parents are more often women [5]. Immigrant women who are single parents have lower social supports compared to Canadian-born counterparts [12]. Not only is women’s health potentially affected [13], but also that of their children. A component of supporting migrant women is to ensure access to health care. The socioeconomic contexts of women migrants need further investigation to better understand how migration intersects with accessing health care. Multiple studies focus on the health and health care delivery needs of refugees but little research focuses on economic migrants and specifically on female economic migrants [14,15].

The purpose of this study was to explore the intersection of place, social connectedness, income, culture, and access to health care and specifically the experiences and perceptions of female economic migrants in Canada from three different African countries. To address the purpose of our research, we include women migrants residing in Canada from countries with different economic classifications: a low–middle-income country (Ghana), a middle-income country (Nigeria), and a middle–high-income (South Africa) country. All three countries are former British colonies and English is one of the official languages spoken in these countries. Economic migrants are selected for their skills and ability to contribute to Canada’s economy. Additionally, more female migrants in Canada are employed than migrant men [6]. There are several subcategories of economic migrants, including skilled workers, business migrants, provincial or territorial nominees, live-in caregivers, and the Canadian experience class [16].

## 2. Methods

We employed a focused ethnography design. The intersectionality approach guided our investigation. Intersectionality refers to the interactions between categories such as gender, race, and other aspects of identity that shape individual lives, as well as social practices, institutional arrangements, and cultural ideologies and outcomes [17]. The central point of the intersectionality approach is the focus on people with different historical backgrounds and experiences of marginalization—in our case, women migrants from three different sub-Saharan African countries, Ghana, Nigeria, and South Africa, with different histories of colonialism and oppression. The intersectionality approach sees social identity as multiple and intersecting. Multiple social identities at the micro level intersect with macro-level structural factors (i.e., poverty, racism, and policies in place) to produce unequal health outcomes [18]. This approach is well suited for research related to understanding how the intersection of social locations affects individuals, their families, and the community.

Ethics approval was received from the Human Research Ethics Review Board at the University of Alberta (Pro-00075943), Canada. Participants were informed in detail about the research project including the benefits and risks of participation. All research staff signed a confidentiality agreement that explained that all data are confidential and private as well as detailed their role in maintaining confidentiality. All participants signed a consent form. Participants had the option to withdraw from the study at any time.

Data were collected over one year. We recruited 29 women from three African countries: Ghana, Nigeria, and South Africa. We used purpose and convenient sampling techniques. Our inclusion criteria were female migrants from Ghana, Nigeria, and South Africa that have migrated; female migrants that classified as an economic class; female migrants that are willing to participate; and female migrants that are in their economic active years of age. We conducted individual narrative interviews to gather female migrant stories and thoughts on place, social connectedness, economics, and culture and how health intersects with these constructs. The format for the interview was informal, with guiding questions developed collaboratively by the research team. This method left room for participants to tell us what they thought was most important. The interviews were face-to-face interviews at a time and place of their convenience. Interviews were audio-recorded and transcribed verbatim. Interviews lasted between 40 and 120 min. Data were analyzed based on thematic content [19] with the support of ATLAS Ti, a computer-based qualitative software for data management. Data were coded by reviewing all interview data and examined for patterns of what was said. Codes were formulated through a line-by-line analysis of concepts identified in the data. Comparative analysis of concepts and participants’ use of concept led to the development of categories—for example, we examined concepts in each transcript and then compared it to other transcripts to determine which concepts were also present. Concepts were then labeled as categories. Themes were developed from the categories that emerged from the data and by comparing them to concepts reported in the literature. Our theoretical approach guided the development of themes to ensure that we addressed the purpose of the study. Rigour was maintained by ensuring the research process was transparent by way of an audit trail, member checking and reflexivity, and ongoing discussion with the research team.

## 3. Findings

We interviewed 10 women from both South Africa and Ghana and nine from Nigeria. Their ages ranged between 24 and 64 years. At the time of the interviews, the majority of the participants had lived in Canada between five months and ten years (*n* = 15), 11 and 15 years (*n* = 5) and more than 15 years (*n* = 9). They were all professional women, with professions that included a social worker, registered nurses, physicians, a child and youth counselor, human resource managers, and a bank assistant. The themes that developed included social connectedness to navigate access to care, place of origin’s influence on access to care, experiences of financial accessibility, and historical and cultural orientation to accessing health care.

### 3.1. Social Connectedness to Navigate Access to Care

Participants shared that social connectedness was important to navigate access to care. Participants reflected on how the lack of being socially connected or the lack of someone who understands the context from where they came affected accessing health care. A South African participant who immigrated with no social connections or family in Canada explained how she had to navigate the system:
“I had no idea how it works. I think, I just went on Google and tried to find a doctor and make the appointment, I didn’t know you had to have, like, coverage for your medications, so that I didn’t know. So, the doctor was gracious enough at that point to give me, like, sample medications, because we didn’t know that you needed medical benefits.”

Another South African participant needed psychiatric care and was *“fortunate again to have had a good colleague who was also [her] family doctor. He said, “‘no you can’t do it like this,’ so … I ended up in his psych unit was…”* She continued by saying she was fortunate to be supported by a colleague:
“What I like health-wise is access to health care was not really a problem for me because I always was fortunate enough to have had a colleague who agreed to be my family doctor, so I never really had a problem with accessing health care but it is a bit awkward if you’re in a group and then that one person has to know everything about your health and I think I was maybe just fortunate to have had very wonderful colleagues who kept it strictly professional.”

A Ghanaian participant shared that *“having someone [from your country] to direct you, someone who understands what is happening”* is very important. She continued by adding how *“important it is to stay connected to a church”* to find a community that is willing to help you to navigate the system. A Nigerian participant elaborated:
“They just embraced us and accepted us into the church family and there was a lady there who, you know, took us under her wing and was just like a big sister to me.”

Some participants used elders from their communities for support to access care. A Ghanaian participant added:
“I selected few elderly people that are from my country and I tell them my problem.”

The nature and sensitivity of the illness experienced by participants influenced the use of social connections, as described by a South African participant:
“I didn’t reach out for help, I guess because I just, how do I explain it, you know, how do, firstly we didn’t have a family doctor, so I didn’t know where to go and I guess I just felt kind of stupid explaining to somebody, you know, okay I’m sad, help me type of thing.”

Participants that were socially connected via family members found it easier to access care. A Nigerian participant said:
“I think my experience was very positive because once we came into Montreal [pause] my husband was told what to do to get his social security number, and I think right away we were hooked up with—I don’t recall the name of the authority now but, you know, with the health authority in Montreal and, yeah, all of the paperwork came with explanations as to, you know, where to go if you needed to see someone.”

Technical health literacy helped participants to access care or get connected, as shared by a Nigerian participant:
“Well, I guess I went onto the internet and Googled, you know, how does one find a doctor and I signed up on, I can’t even remember the name of it, it was some kind of health connect. And you had to fill in forms and whatnot and then we got contacted after a few months …”

### 3.2. Place of Origin Influence on Access to Care

The place of origin in the context of this study refers to the country the participants migrated from. It was apparent how the health care system in the country of their origin influenced how they experienced access to care in Canada. A participant shared that the differences in how health care is delivered and the accessibility to her physician in Canada affected her access to care:
“… in South Africa, we were so privileged to be part of the system where we had health coverage, so you had your personal physician that you could access much easier than here. What I find difficult here is that I have a doctor here that I go to once a year for my annual visits, but when you’re sick and you phone his office, they will tell you, you can [only] get an appointment next week… so you have to go to a walk-in clinic to a doctor that don’t know you, they don’t know your history. I find, I don’t like that at all, because I need to have a personal relationship with my physician…So, I will now really postpone to go and see a person at a clinic, so I will usually when I go to a clinic, is when I’m really, really, really sick, that it’s bad, because I know I should’ve gone already much earlier. So, in that sense, I think access to health care is not too bad here, it’s available, but that personal relationship that I had with my physician in South Africa was different.”

The system in Canada, where you have to be accepted by a doctor, influenced the access to care of new migrants. A Nigerian participant shared:
“That’s correct, yes. Like, everybody was like, when you phone him, ‘We don’t accept new patients.’ I’m like, what if you have this critical condition? Or, if you had an emergency, what do you do?”

A Ghanaian participant shared that her access to preventative medicine was much better in Canada than in her country of origin:
“So, for instance, things like PAP smear and it wasn’t there. But, here, it’s part of your routine. So that’s why a lot of people by the time they realize they have an illness, it’s too late.”

A South African participant added:
“And, you get the flu shots and, there’s no cost. And, you go for your yearly checkups and, there’s no cost.”

Participants shared that the wait time in Canada is a factor in access to care. A South African participant shared:
“Otherwise, I think health-wise the positive is, I know everybody’s complaining about the wait times, but you get the services that you need. Like the mental health, I received the mental health services. If I need to go and see a psychologist, I’m able to do that. If there is any mental medical tests that needs to be done, I know those options are here and it’s available to us. So, that’s all positives.”

Another Nigerian participant added:
“… it’s just the waiting times, I think, to get to doctors. Some doctors, that you have to wait months to see them. That’s the only thing that’s frustrating here, is getting a doctor and getting into a doctor, like, a specialist, kind of thing.”

### 3.3. Experiences of Financial Accessibility

Participants had positive views on financial access to care in Canada in comparison to their countries of origin. All the participants shared that access to health care in their countries of origin was more expensive than in Canada. A South African participant shared that health care in Canada is “is probably better because … it’s more equitable for society because people don’t have to pay for their primary care.” A Nigerian added about her positive experience:
“I should say it’s affected my health positively. Yeah, maybe positively. Because I’m able to have this contact with my physician, she’s did so many tests. Back home I would have to pay a lot for me to go for any studies or diagnostic tests and all worth not, but here, once you present your health card you are covered with whatever. So even something that you might be suspecting that might be happening to you, just go for a test and it works. Like there’s not so much stress about it. But maybe back home, because you don’t have the money, you tend to procrastinate or postponing it until it damages your system, stuff like that. But it’s not so here. Even if I’m suspecting something is happening, it’s just for me to walk in there and then and they attend to me. So, I think it’s positive.”

A Ghanaian participant had the same experience. She elaborated:
“Ghanaians don’t take care of their health, because even now it’s even worse because they pay, you know… When you go to the main government hospitals, you don’t pay anything. But now they pay for things and there’s more focus on health here than there is in Ghana.”

Another Nigerian participant added:
“Health for me means being able to go to the hospital when you are sick and getting attention, getting medical help anytime you are sick and not thinking about the money. Comparing it to Nigeria, you know, you do not go to the hospital unless you have money, because you have to pay for the card and you have to pay for hospital bills. So, lots of people can’t go to the hospital because they can’t afford it. But in Canada, because you … they take it from if you work and you have health insurance you do not worry about that.”

However, while this participant was satisfied with the services, a South African participant elaborated about the out-of-pocket costs for services such as dental and eye care:
“Yeah, I think health care, general health care was good but little things like both my girls had teeth problem that they have to straighten their teeth and all of that kind of stuff and that was really a tough one because it was very expensive to do that. Things like glasses, because we all wear glasses, glasses, teeth, you know those special little things. That was a tough one in the beginning, until we could get on a good medical aid, private medical aid to do that, but it’s still they don’t cover a lot of these little extra things. So that was a hard one. I mean, still, even today, if you want to go to a dentist and you have to do a lot of dental work it’s hard. It’s not easy to do that because you have to pay so much out of your own pocket for it.”

### 3.4. Historical Cultural Orientation to Accessing Health Care

The culture of participants and how the health delivery system adapts to migrants from different countries affect access to care. A Ghanaian participant talked about the cultural components of care:
“Medicine, no. No, that was not even the issues, when you come in here, our traditional medicine all have connotation of some spiritual something. So, a Ghanaian woman or a Ghanaian man will be very happy when their doctor talk to you and said, oh you have this, this, this, this and this and you have to take this medicine and then we have to pray. That word pray gives them the placebo effect, even if they give them water, they know the doctor is going to pray, so they’re going to be okay. Because everything that we do with medicine has connotation with praying.”

She continued by explaining that the health care workers have to spend more time with migrants to explain the illness process and the use of the treatment:
“To adapt, it’s a gradual process, it’s a gradual process, you need somebody to talk to you, what the procedure is like and what is accepted and what is not accepted. And then one other thing is because we do the herbs at home, maybe you drink, maybe it’ll be three of these cups and you are okay. So, when you give them the medicine and they two, three days and then they feel okay, then they stop, but they are not healed, then they. So, the whole things keep coming over and over and over and over again. Where in Ghana, when you cook all your herbs and drink it, one or two, yeah okay, you are okay, which is not here … My advice would be is as soon as they come, they have to explain things to them, how it works here and the difference between what works here, what works home, might not work here, you know. Because we know some parents who when they came, they didn’t want to give their children any European medicine or the conventional medicine, so they are trying to do that. And some, a long time ago, some people killed their children, you know, like the child have meningitis, you don’t take the children to hospital and then you said, ‘oh, ginger and lemon’ and then by the time they rush to the hospital, it’s too late.”

A Nigerian participant elaborated on the difference in cultural ways of accessing care:
“… back home there is lots of self-medications. Here, I brought all the meds I could. Back home there is a codeine problem … they drink it to get high. They closed pharmacists who provided it back home. Now tramadol. Abortion in Nigeria is not allowed. So, people give themselves abortions and take medications for it. They can get it over the counter … it’s so easy to get the strongest of drugs over there but not here. In the first two weeks, we got health cards for everyone. I want a doctor who understand African pathology.”

A South African participant shared that for her there was not much difference in accessing care:
“I think the quality here is excellent, I was also so privileged to do, like I’ve said, already work in other African countries, so I know what the quality of health care is there. So, I think as Canadians, we have absolutely no reason to complain about any health care quality. I think the standards are good, like I said, I would’ve loved more direct access to my own doctor, but that’s more or less my only complaint.”

The next quote highlights how different factors intersect with access to health care. A Nigerian participant shared:
“It’s a huge mountain that if you don’t have the necessary info, they keep turning you around back and forth. Back home if there is an emergency, you take your child and go. Here, you call 911. Back home as long as you have money you get services. But here if my son is having problem I have no idea where to go. Do I go to [children’s hospital] or [name of another hospital]? I hope nothing happens with them. Back home there is lots of self-medications…Back home during flu season I don’t get kids immunized. I make sure they drink more water and eat vitamin C more. Here, we got the flu shots, but I want a doctor from Nigeria who understands our own pathology more.”

## 4. Discussion

Many complex and intersecting factors affect access to health care; these factors are also often unknown. Our study explores the intersections between migration and access to health care of economic female migrants. It was clear that social connectedness, place of origin, income and financial accessibility, and historical cultural orientation to access health care affected economic migrant women’s access to health care after migration. Canada has a universal health care system but multiple research studies have documented that migrants have significant barriers to accessing health care [20]. Most migrants indeed arrive in Canada from a health care system that is very different than their country of origin. Access to health care is one of the most important social determinants of health.

Individual lifestyle factors, social and community connections, social economics, and the cultural and environmental context contribute to new migrants’ integration into a new society [21]. Most people migrate for a better quality of life but, irrespective of their immigrant status, it is not without challenges. Migrants often report poorer health than the local populations and poorer health behaviors and quality of life [22]. The ‘healthy immigrant effect’ is well known in the Canadian context, where migrants arrive relatively healthy in Canada, and then their health declines over time [23]. Access to health care from a global perspective is important. The United Nations Sustainable Goals are the blueprint to ‘leaving no one behind’ [24]. Goal 3 and 5 of the Sustainable Development Goals, Good Health, and Well-Being, and Gender Equality, state clearly that access to health care and giving women equal access to health care are the most important components to achieving universal health care [3] and health equity for all. Access to care intersects with race, ethnicity, socioeconomic status, age, sex, sexual orientation, gender identity, and residential location [25].

It is important to look at migration from a gendered perspective, hence our focus on economic female migrants. Much has been written on migration and integration, but not much attention has been given to the vulnerability of women in the migration process and *“their potential as key actors for integration. Women may play this role by promoting the integration of their families and social circles, supporting their children’s education,”* and accessing health care for their family [26] (p. 1). Additionally, knowledge about the quality and efficacy of this access to care is not well researched, especially for migrant women [27].

Financial accessibility was a factor highlighted by our participants. Our participants were economic migrants, it was therefore surprising that ohers factors related to access to care such social connectedness and their comfort with historical experiences with the health care system in their countries of origin, were also at play. Levesque, Harris, and Grant Russell [28] conceptualize five intersecting dimensions of accessibility of services that apply to our findings. These dimensions include approachability, acceptability, availability and accommodation, affordability, and appropriateness. Additionally, each dimension is influenced by the equivalent abilities of persons and, in our case, economic migrant women interact with the dimensions of accessibility to create access to health care. These dimensions include the ability to perceive, seek, reach, pay, and engage.

Our findings indicate that health beliefs and health literacy influenced our participants’ ability to perceive. The lack of social connectedness intersected with their perceived ability to access and seek information. Women’s ability to seek health care includes interventions that comprise community connections and involvement activities [29]. *“Health literacy becomes a means for not only acquiring knowledge but also as a resource for engaging in health at personal and community levels”* [30] (p. 2). The development of critical health literacy skills is important to improve immigrant women’s access to health care. *“Critical literacy is based on the idea that language constructs the lenses we use to make sense of the world”* [31] (p. 1). The lenses women are using to make sense of the world include the intersection of cultural and gender values. These influence our participants’ ability to seek and the lack of social support influences their ability to reach for health care. The ability to pay, for example, for dental care for their children, influenced our participants’ access to using these services, and the ability to engage in health care was influenced by a lack of information and caregiver support.

Acculturation to the health care system in Canada is additionally influenced by the historical cultural orientation related to countries of origin. The participants reflected on the acceptability of services and particularly on the lack of holistic care and were searching for health care providers from their countries of origin. Creating suitable educational and management systems, which include the promotion of conscientiousness and supporting holistic care will ultimately improve the quality of access to health care for immigrant women [32].

Recommendations to improve the access to care of economic migrant women include a variety of suggestions. Community navigators are also known as cultural brokers, *“community health workers, outreach workers, promotors, lay health educators, health advocates, peer counselors or medical assistants”* [33] (p. 1) and have been shown to reduce existing barriers to accessing health care. Shommu et al. [34] conducted a systematic scoping review on the use of community navigators that offers *“culturally tailored educational support” to newcomers and “guide patients in overcoming barriers to obtaining appropriate health care”* [34] (p. 2). The review concludes that culturally competent guidance is very efficient to support access to care.

Structural barriers to the health care system pose different access challenges [33]. Improving access to and the quality of health care for migrants is a primary responsibility of health systems. *“An effective solution would be the adoption of a ‘whole organizational approach’”* that develops *“specific programs that address priorities”* for migrants including processes and services as well as training of health care providers to deliver culturally appropriate care that is coordinated and equitable [34]. Many community-based programs are focusing on migrants with low educational levels and who have language barriers. These programs can be adapted to address the specific needs of economic female migrants.

## 5. Conclusions

Worldwide, a variety of complex and intersecting factors affect the ability of health care systems to provide accessible health care. Our study focused on intersecting factors that influence female economic migrant women’s ability to access the health care system. Economic female migrants face challenges that uniquely intersect with gender roles within their families. They experience stressors related to caring for their extended family members that were left behind, while adjusting to the new culture in Canada, caring for their families and establishing themselves in a new work environment. One of the basic needs of the family is caring for their health needs. Caring for a family’s health care needs and accessing care comprise one of the major stressors for female migrants.

Health care providers play a central role in health systems globally and are key in facilitating access to health care for all, but especially for individuals and communities during vulnerability. Interventions to improve social connectedness with communities with the same social cultural orientation and place of origin will support access to care—for example, sharing a list of physicians from the migrants’ countries of origin. Different options to support financial accessibility will be important—for example, during regular health care visits, the social determinants of health should form part of the regular assessment. This will support alternative solutions if a migrant has financial accessibility challenges.

Access to care is one of the most important aspects that is needed to improve the health of a population—in our case, economic female migrants and their families. Further research will be importantly related to the different levels of health literacy and the needs of different populations of migrants. Our findings have clear policy implications, e.g., the development of integration courses and support groups for immigrant women.

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
