# Peer review of "Intersection of Migration and Access to Health Care: Experiences and Perceptions of Female Economic Migrants in Canada"

_ijerph, 2020, doi:10.3390/ijerph17103682_

Round 1

Reviewer 1 Report

I enjoyed reading the manuscript. Access to healthcare for migrant women is a growing global concern. Any quality evidence informing better practices in this regard is timely and important. The research is original and advances current knowledge. The results are significant and well presented. The study design is appropriate and the analyses robust. 
In my view the results and recommendations will be of interest to a wide readership of this Journal. The writing is professional and understandable. 

Author Response

Dear reviewer

Thank you for the positive review of the manuscript.  We have attended to the suggested minor language editing. 

I am attaching the revised version with the changes required by reviewer 2.  the language editing have been highlighted in Yellow. 

Reviewer 2 Report

This is an interesting article in a number of respects. Generally, it looks at immigrant women from three relatively better off African countries, Ghana, Nigeria, and South Africa, and their adaptation to the Canadian health care system. The 29 women interviewed are all from a professional background, and, in fact, there were physicians, registered nurses and a social worker. This is hardly what one might considered to be non-prestigious occupations. The article attempts to apply an intersectionality approach to its study with a thematic content analysis based on the transcribed interviews of those included in their sample of 10 women each from Ghana and South Africa and nine from Nigeria. The quotations presented to support the four thematic areas identified: social connectedness to navigate access to care; place of origin influence on access to care; experiences of financial accessibility; and, historical cultural orientation to accessing health care, were interesting. The discussion section was interesting and included a number of observations and recommendations for community navigators or cultural brokers to facilitate access to health care and a "whole organizational approach" that would include specific programs to address immigrant priority needs along with training for health care providers to deliver culturally appropriate care that is coordinated and equitable.

Nonetheless, I have some serious concerns regarding this ethnographic study and how it is presented that can be divided into methodological and analysis concerns. With respect to methodology, let me begin by saying that 29 interviews with professional women immigrants, 24 to 64 years of age and that have been in Canada from five months to 31 years, from three African countries, raises serious concerns regarding generalizing findings. It is noted that the average length of time spent in Canada was 12 years. Clearly, the sample drawn upon, as a whole, is not recent immigrants. When does one become fully resettled in their new country and society? How many of these immigrant women were Canadian citizens or permanent residents? Any findings based on only 29 interviews would be exploratory or suggestive at best.

Further, why were the women selected from these three African countries? South Africa is recognized as one of the more affluent countries in Africa that receives thousands of migrants from other parts of Africa. Indeed, the quotes from some of the South African immigrant women seem to suggest that they lost personal connections with their doctors in Canada that they had previously enjoyed in South Africa. Nigeria despite its difficulties is also known as a oil rich and better off country in Africa. Likewise, Ghana is considered to be relative stable and one of the better off countries within Africa that also hosts migrants from neighbouring countries in the region. I believe that it would be fair to provide a rationale for the selection of these three African countries for this study. All three are former British colonies where English is a common language which might suggest that this was one of the reasons that immigrants from these countries were selected.

There was too little explanation of what interview instrument was used for the interviews. What questions were posed to the persons who were interviewed? Were they open ended or closed questions? How long were the interviews? 

How were the four thematic areas identified? Why have you selected these quotations as opposed to others? I found that there was too little information presented regarding the content analysis of the interviews based on the questions that where asked of the female respondents.

With respect to the analysis of your data, there are a number of concerns. First, despite employing an intersectionality approach that recognizes that social identity is multiple and intersecting there was no effort to apply this analysis from the data gathered in the interviews. This could have easily been done on the basis of, for instance, race, age, occupation, citizenship, education, sexual orientation, length of time in Canada, languages spoken, religious background, and so on. Second, what was the basis of the selection of the quotations in each of the four thematic areas identified? Are these intented to be representative of the three groups of women? Here it would have been good also to know whether you found any difference between those among the three groups of women. It could be presumed that there would be a major difference between the women in South Africa and those from West Africa. Likewise, tribal affiliation and/or ethnicity might be an important variable here as well. 

With respect to the Discussion section of the paper, the connection between the review of the literature and the findings from of the study to the observations and recommendations was not clearly evident. The linkages here need to be made more explicit.

Finally, I found the Conclusions to be highly underdeveloped based on the nature of your ethnographic study of immigrant women from three African countries who have resettled in Canada. To conclude with the statement that there should be integration courses developed and support groups for immigrant women is to beg the question what is wrong with those that are in place now?

Additionally, it would have been good to cite the confidentiality agreement at line 92. And, there seems to be an error in the sentence at line 152. 

Author Response

Dear reviewer 

Thank you for the detailed feedback.  I believe the revisions will contribute to the quality of the manuscript. We have listed all your concerns and responded in Italics. I am additionally uploading a new version of the manuscript with the changes highlighted. 

1. This is an interesting article in a number of respects. Generally, it looks at immigrant women from three relatively better off African countries, Ghana, Nigeria, and South Africa, and their adaptation to the Canadian health care system. The 29 women interviewed are all from a professional background, and, in fact, there were physicians, registered nurses and a social worker. This is hardly what one might considered to be non-prestigious occupations. The article attempts to apply an intersectionality approach to its study with a thematic content analysis based on the transcribed interviews of those included in their sample of 10 women each from Ghana and South Africa and nine from Nigeria. The quotations presented to support the four thematic areas identified: social connectedness to navigate access to care; place of origin influence on access to care; experiences of financial accessibility; and, historical cultural orientation to accessing health care, were interesting. The discussion section was interesting and included a number of observations and recommendations for community navigators or cultural brokers to facilitate access to health care and a "whole organizational approach" that would include specific programs to address immigrant priority needs along with training for health care providers to deliver culturally appropriate care that is coordinated and equitable.

Nonetheless, I have some serious concerns regarding this ethnographic study and how it is presented that can be divided into methodological and analysis concerns. With respect to methodology, let me begin by saying that 29 interviews with professional women immigrants, 24 to 64 years of age and that have been in Canada from five months to 31 years, from three African countries, raises serious concerns regarding generalizing findings. It is noted that the average length of time spent in Canada was 12 years. Clearly, the sample drawn upon, as a whole, is not recent immigrants. When does one become fully resettled in their new country and society? How many of these immigrant women were Canadian citizens or permanent residents? Any findings based on only 29 interviews would be exploratory or suggestive at best.

Author response.  We understand your remark about the sample size and composition.  The recruitment criteria were purposive sampling and inclusion criteria in the original proposal clearly stated:

Our inclusion criteria are as follow:

  • female migrants from Ghana, Nigeria, and South Africa that have migrated,
  • female migrants that classified as economic class,
  • female migrants that are willing to participate, and
  • female migrants that are in their economic active years of age.

The sample fitted within these criteria. The purpose is not to generalize but to develop an in-depth understanding of the phenomenon. The number of participants is in line with other ethnographic studies.  We collected data until we could present an in-depth and rich description of the phenomenon. 

We have not collected data on the citizen status in Canada and will therefore not be able to reflect on that.

We have added a clear section (“Their ages ranged between 24 to 64 years. At the time of the interviews, the majority of the participants had lived in Canada between five months and ten years (n= 15), and 11 to 15 years (n=5) and the rest more than 15 years (n= 9)). on the length of stay in Canada.  Only small about of the participants have stayed for a longer period in Canada. 

2. Further, why were the women selected from these three African countries? South Africa is recognized as one of the more affluent countries in Africa that receives thousands of migrants from other parts of Africa. Indeed, the quotes from some of the South African immigrant women seem to suggest that they lost personal connections with their doctors in Canada that they had previously enjoyed in South Africa. Nigeria despite its difficulties is also known as a oil rich and better off country in Africa. Likewise, Ghana is considered to be relative stable and one of the better off countries within Africa that also hosts migrants from neighbouring countries in the region. I believe that it would be fair to provide a rationale for the selection of these three African countries for this study. All three are former British colonies where English is a common language which might suggest that this was one of the reasons that immigrants from these countries were selected.

Author response:  A stronger rational was added. See line 74 – 77.  A further stong rationla can be find in lines 88 – 90.  

There was too little explanation of what interview instrument was used for the interviews. What questions were posed to the persons who were interviewed? Were they open ended or closed questions? How long were the interviews? 

Author response: We have included the following information related to the interview instrument:

We conducted individual narrative interviews to gather female migrant’s stories and thoughts on how place, social connectedness, economics, culture, and health intersects. The format for the interview was informal with guiding questions developed collaboratively by the research team. This method left room for participants to tell us what they thought was most important. The interviews were conducted face-to-face interviews at a time and place of their convenience. Interviews were audio-recorded and transcribed verbatim. Interviews lasted between 40 – 120 minutes.  

3. How were the four thematic areas identified? Why have you selected these quotations as opposed to others? I found that there was too little information presented regarding the content analysis of the interviews based on the questions that where asked of the female respondents. 

Author response: I am quoting what is currently in the manuscript: “ Data were analyzed based on thematic content [18] with the support of ATLAS Ti, a computer-based qualitative software for data management. Data were coded by reviewing all interview data and examined for patterns of what was said. Codes were formulated through a line-by-line analysis of concepts identified in the data. Comparative analysis of codes and participants’ use of codes led to the development of categories. Themes were developed from the categories that emerged from the data and by comparing to concepts reported in the literature. Our theoretical approach guided the development of themes to ensure that we answered to the purpose of the study. Rigour was maintained by ensuring the research process was transparent by way of an audit trail, member checking and reflexivity, and ongoing discussion with the research team.”

I am not sure how we can be more clear without becoming too wordy.

The quotes that were selected were based on how ell it presented the theme/subtheme and we tried to represent participants from the 3 countries.

4. With respect to the analysis of your data, there are a number of concerns. First, despite employing an intersectionality approach that recognizes that social identity is multiple and intersecting there was no effort to apply this analysis from the data gathered in the interviews. This could have easily been done on the basis of, for instance, race, age, occupation, citizenship, education, sexual orientation, length of time in Canada, languages spoken, religious background, and so on. Second, what was the basis of the selection of the quotations in each of the four thematic areas identified? Are these intented to be representative of the three groups of women? Here it would have been good also to know whether you found any difference between those among the three groups of women. It could be presumed that there would be a major difference between the women in South Africa and those from West Africa. Likewise, tribal affiliation and/or ethnicity might be an important variable here as well. 

Author response: We appreciate this feedback.  The specific purpose of this manuscript was to compare participants from the different countries,  That said we are in the process of writing other manuscripts where will talk about differences. 

5. With respect to the Discussion section of the paper, the connection between the review of the literature and the findings from of the study to the observations and recommendations was not clearly evident. The linkages here need to be made more explicit.

Author response: We have rearranged the content and added some sentences to make the connection between the literature and the findings more clear. I have additionally highlighted the areas to show the connections.

Finally, I found the Conclusions to be highly underdeveloped based on the nature of your ethnographic study of immigrant women from three African countries who have resettled in Canada. To conclude with the statement that there should be integration courses developed and support groups for immigrant women is to beg the question what is wrong with those that are in place now?

Authors response:   Very good remark. Many of the current programs are developed for the refugee population and immigrant groups that have language barriers.  Not much is developed for economic migrants.  We have added two sentences at the end of the discussions.   We have reworked the conclusion.

Additionally, it would have been good to cite the confidentiality agreement at line 92. And, there seems to be an error in the sentence at line 152.

Author response: Sentence 152 was corrected.

The confidentiality agreement was an agreement that was approved by the Human Research Ethics Review Board and not an official developed one that I can refer to.  It is common practice to develop an individual confidentiality agreement and it is then approved.  It might differ from study to study depending on the nature of the study and who need access to the data.

Round 2

Reviewer 2 Report

Thank you for these revisions. I do not consider them major but minor with points clarified here and there.

I still have a number of outstanding concerns with this paper.

-- The conflation of migrant with immigrants. The use of the word "migrant" at times recognizes there are immigrants, that also are referred to at times as economic immigrants, and that there are migrants and refugees. Migrants encompass the broad category of all those who travel from their countries of origin to their new countries, sometimes permanently. This includes immigrants, regardless of their motivation to resettle elsewhere, in other words, voluntary migrants, and refugees, involuntary migrants, who are forced to leave their countries of origin, then, there are those who travel for education, vacation, adventure, religious purposes, and, there are those who are illegal and may be in hiding for various reasons, for example, criminality, that are referred to as irregular migrants. 

-- Your study is unique because it deals with voluntary migrants who have professional backgrounds and have come from three countries: Ghana, Nigeria, and, South Africa. The assumption you seem to make is that they came to Canada for economic reasons; that is, a better quality of life.

-- You make a broad assumption that the subjects of your study are vulnerable persons because they immigrated from abroad. Given their professional backgrounds and the length of the time they have resided and worked in Canada this seems to run contrary to your broad assumption of vulnerability. They certainly do not fall in the category of refugees, persons who fled their countries due a well-founded fear of persecution and may have experience severe trauma prior to their arrival in Canada seeking asylum.

-- The connection between the questions asked of the respondents of your study and the data coding and analysis and the formulation of your four thematic categories and the selection of quotes from respondents of these three countries is not sufficiently explicit. For instance, what does the following sentence mean at line 114? "Comparative analysis of codes and participants' use of codes led to the development of categories." This may be clear to you but it is not at all clear to the reader.

-- The Conclusions are improved but what do they tell us? That immigrant women's access to health care is contingent on at least the four variables that you have identified: social connectedness; place of origin; financial means; and, historical cultural orientation. But, you end with stating the policy implications are integration courses and support groups for immigrant women. Shouldn't it somehow be related to your four variables and their interactive and compounding effects? For instance, from your own quotes, the respondents wanted to have physicians who were familiar with their own cultural pathologies and remedies. Not having comprehensive medical coverage for dental treatments and glasses made it financially difficult for respondents with children, and so on. I would encourage you to consider what the respondents have stated and not to generalize to the extent that it takes you beyond your own data.

There are writing errors at the following lines:

102; 108; 163; 331-2; and 385.

Your abstract should include your major findings, not simply a listing of the four variables, but, also your conclusions.  

Author Response

May 15th, 2020 response to reviewer

Reviewer: The conflation of migrant with immigrants. The use of the word "migrant" at times recognizes there are immigrants, that also are referred to at times as economic immigrants, and that there are migrants and refugees. Migrants encompass the broad category of all those who travel from their countries of origin to their new countries, sometimes permanently. This includes immigrants, regardless of their motivation to resettle elsewhere, in other words, voluntary migrants, and refugees, involuntary migrants, who are forced to leave their countries of origin, then, there are those who travel for education, vacation, adventure, religious purposes, and, there are those who are illegal and may be in hiding for various reasons, for example, criminality, that are referred to as irregular migrants. 

Response:  I have re-read the manuscript to ensure that I consistently use the term ‘migrants’ and that it is clear that this study focuses on migrant women that have to migrate for economic reasons. I have removed all references to refugees.

Your study is unique because it deals with voluntary migrants who have professional backgrounds and have come from three countries: Ghana, Nigeria, and, South Africa. The assumption you seem to make is that they came to Canada for economic reasons; that is, a better quality of life.

You make a broad assumption that the subjects of your study are vulnerable persons because they immigrated from abroad. Given their professional backgrounds and the length of the time they have resided and worked in Canada this seems to run contrary to your broad assumption of vulnerability. They certainly do not fall in the category of refugees, persons who fled their countries due a well-founded fear of persecution and may have experience severe trauma prior to their arrival in Canada seeking asylum.

Response: I have added a sentence to indicate that all migrants are exposed to transition and that the transition exposes migrants to different levels of vulnerability. I have added a reference. Subsequently, the numbers for the references in the text have been adapted.

The connection between the questions asked of the respondents of your study and the data coding and analysis and the formulation of your four thematic categories and the selection of quotes from respondents of these three countries is not sufficiently explicit. For instance, what does the following sentence mean at line 114? "Comparative analysis of codes and participants' use of codes led to the development of categories." This may be clear to you but it is not at all clear to the reader.

Response: I have added a better description of what comparative analysis means.

The Conclusions are improved but what do they tell us? That immigrant women's access to health care is contingent on at least the four variables that you have identified: social connectedness; place of origin; financial means; and, historical cultural orientation. But, you end with stating the policy implications are integration courses and support groups for immigrant women. Shouldn't it somehow be related to your four variables and their interactive and compounding effects? For instance, from your own quotes, the respondents wanted to have physicians who were familiar with their own cultural pathologies and remedies. Not having comprehensive medical coverage for dental treatments and glasses made it financially difficult for respondents with children, and so on. I would encourage you to consider what the respondents have stated and not to generalize to the extent that it takes you beyond your own data.

Responds:  I have added a section that related more to the specific findings and hopes it is clearer.

There are writing errors at the following lines:

102; 108; 163; 331-2; and 385.

Your abstract should include your major findings, not simply a listing of the four variables, but, also your conclusions.  

Response: The writing errors have been attended too. It difficult to add a discussion of the major findings in the abstract as I am restricted by a minimum amount of words. I have, however added a short conclusion.
